# The Emerging Function of PKCtheta in Cancer

**DOI:** 10.3390/biom11020221

**Published:** 2021-02-05

**Authors:** Amandine Nicolle, Ye Zhang, Karine Belguise

**Affiliations:** MCD, Centre de Biologie Intégrative (CBI), Université de Toulouse, CNRS, UPS, 31062 Toulouse, France; amandine.nicolle@univ-tlse3.fr (A.N.); ye.zhang@univ-tlse3.fr (Y.Z.)

**Keywords:** PKCtheta, cancer, tumoral function, mechanisms of action

## Abstract

Protein Kinase C theta (PKCθ) is a serine/threonine kinase that belongs to the novel PKC subfamily. In normal tissue, its expression is restricted to skeletal muscle cells, platelets and T lymphocytes in which PKCθ controls several essential cellular processes such as survival, proliferation and differentiation. Particularly, PKCθ has been extensively studied for its role in the immune system where its translocation to the immunological synapse plays a critical role in T cell activation. Beyond its physiological role in immune responses, increasing evidence implicates PKCθ in the pathology of various diseases, especially autoimmune disorders and cancers. In this review, we discuss the implication of PKCθ in various types of cancers and the PKCθ-mediated signaling events controlling cancer initiation and progression. In these types of cancers, the high PKCθ expression leads to aberrant cell proliferation, migration and invasion resulting in malignant phenotype. The recent development and application of PKCθ inhibitors in the context of autoimmune diseases could benefit the emergence of treatment for cancers in which PKCθ has been implicated.

## 1. Introduction

The Protein Kinase C (PKC) family is a family of serine/threonine kinases that are involved in various cellular processes for different cell types. The PKC family is classified into three subfamilies: classical (α, βI, βII, γ), novel (δ, ε, η, θ) and atypical (ζ, ι/λ) PKC isoforms. This classification is based on their structure and ability to respond to calcium and/or diacylglycerol (DAG) [1]. Among this family, the novel PKCθ isoform is different from other PKC isoforms since its physiological expression is limited to a few types of cells, such as T cells, platelets and skeletal muscle cells. This specific expression confers to this isoform a central role in the immune system where PKCθ controls T cell activation, survival and differentiation [2]. In skeletal muscle cells, PKCθ regulates muscle cell development, homeostasis and remodeling [3]. Beyond its physiological functions, PKCθ is also involved in the pathology of various diseases. In the context of the immune system and skeletal muscle tissue, the dysregulation of PKCθ activity leads to both autoimmune and inflammatory diseases and to insulin resistance and Type 2 diabetes, respectively [3,4]. In the last decade, growing evidence implicated the PKCθ signaling in the biology of cancer where it controls cancer cell proliferation, migration and invasion at the cytoplasmic or nuclear levels. Here, we discuss this emerging function of PKCθ in cancer by analyzing its diverse modes of action and their consequence on critical biological processes involved in tumorigenesis and cancer progression.

## 2. PKCθ Structure and Physiological Function

In this section, we provide a brief overview of the PKCθ structure and the PKCθ physiological function mainly in the immune system. For extensive details, the readers can refer to several excellent reviews written by the experts in the field of T cell biology (reviewed in [2,4,5,6,7]).

### 2.1. PKCθ Structure

The novel PKCθ isoform is a protein kinase encoded by the *PRKCQ* gene and composed of 706 amino acids with a molecular weight of approximately 82 kDa [8]. PKCθ is a DAG-dependent but Ca^2+^-independent, protein kinase whose structure consists of several functional domains that are conserved among the novel PKC subfamily (Figure 1) [1]. The N-terminal regulatory domain contains the C2-like domain, the pseudosubstrate region and the DAG-binding domain (C1A/B) while the C-terminal catalytic domain contains the ATP-binding domain (C3) and the substrate-binding domain (C4). The regulatory and catalytic domains are separated by a hinge region, called the V3 motif, which is unique and highly specific to each PKC isoforms.

### 2.2. PKCθ Function in the Immune System

Due to the high expression levels of PKCθ in T lymphocytes, extensive research has studied the biological function of this novel PKC isoform in the immune system. The generation and analysis of PKCθ-deficient mice have unraveled the selective role of PKCθ in the T cell immune response [9,10]. While PKCθ is critical for the T helper (Th)2- and Th17-mediated responses, the Th1- and cytotoxic T cell-driven responses remain relatively intact in the absence of PKCθ [4,7]. However, a few studies reported that some specific Th1 responses were altered in PKCθ deficient mice [11,12]. T lymphocyte activation is a central step of the T cell immune response during which T cell interacts with an antigen-presenting cell (APC) [4]. This cell-cell junction forms a well-organized and dynamic structure called the immunological synapse [13]. Following this T cell-APC interaction, cytoplasmic PKCθ is translocated to the membrane at the immunological synapse [6] and this specific and critical relocalization is highly dependent on the unique V3 motif of PKCθ [14]. In addition, other events are also required for the proper localization and activation of PKCθ at the immunological synapse. Concerning the PKCθ localization, several studies indicated that the lck-mediated phosphorylation of PKCθ at tyr-90 participated in the PKCθ recruitment to the immunological synapse [14,15] and a report from Thuille et al. suggested that the PKCθ autophosphorylation at thr-219 was required for the cell membrane localization of PKCθ [16]. Moreover, the data from Cartwright et al. suggested that PKCθ required its active kinase domain in order to be maintained at the immunological synapse [17]. More recently, Wang et al. reported that the sumoylation of PKCθ upon T cell activation was involved in the specific localization of PKCθ and in the organization of the immunological synapse [18]. Concerning the PKCθ activation, the phosphorylation at Thr-538 in the activation loop regulates the PKCθ activity by maintaining PKCθ in an active conformation and thus this phosphorylation has been used as a marker reflecting the PKCθ activation [19]. GCK-like kinase (GLK, MAP4K3) has been identified as one kinase capable of directly phosphorylating this Thr-538 residue during the T cell activation [20]. Moreover, the auto-phosphorylation at Ser-695 induced during T cell activation is also required for the PKCθ kinase activity [19,21]. 

Once translocated to the immunological synapse, PKCθ integrates various signaling cascades that conduct to the activation of important transcription factors, including Nuclear Factor κB (NF-κB), Activating Protein 1 (AP-1) and, to a lesser extent, Nuclear Factor of Activated T-cells (NFAT) [5]. This transcriptional machinery then induces the production of interleukin-2, a cytokine essential for the T cell proliferation [5]. Moreover, the PKCθ function is not only limited to the activation of signaling pathways that leads to the transcriptional regulation of gene expression. For example, PKCθ has been involved in the actin cytoskeletal reorganization that occurs during the formation of the immunological synapse and the related polarization of activated T cells [18,22,23]. PKCθ can also enter the nucleus of activated T cells to directly bind to the chromatin in order to regulate the expression of immune response genes and microRNAs involved in the cytokine regulation [24].

### 2.3. Implication of PKCθ in Immunological Disorders

As a selective regulator of the Th2 and Th17 immune responses, the perturbation of PKCθ expression and activity leads to the development of Th2-driven inflammatory diseases and Th17-mediated autoimmune diseases. Indeed, PKCθ is highly expressed and activated in these immunological disorders [4]. Studies from the PKCθ-deficient mice showed that the PKCθ suppression decreased the T cell inflammatory response in autoimmunity, allergy and allograft rejection [4]. Therefore, the therapeutic use of specific PKCθ inhibitors could provide an interesting approach for these PKCθ-dependent pathologies [25]. Clinical studies using sotrastaurin (AEB071) as the PKCθ inhibitor showed some encouraging results in the context of immunosuppressive therapy for autoimmune diseases such as psoriasis and organ transplantation [4,26]. However, sotrastaurin is not specific to PKCθ and it also shows strong and specific inhibitory activity against PKCα and PKCβ and to a lesser extend against PKCδ, PKCε and PKCη. It thus suggests that sotrastaurin would inhibit not only the PKCθ-mediated functions but also the functions from other PKCs [27]. Therefore, current research works aim to develop more selective PKCθ inhibitors [28,29]. These inhibitors are currently tested in mouse models and further studies are needed to validate them in the clinical trials.

## 3. PKCθ Function in Cancer

### 3.1. Leukemia

PKCθ has been proved to contribute a lot to the proliferation, activation and survival of cytotoxic lymphocytes and leukemic T cells. Using wild-type (WT) and PKCθ-deficient mice with Moloney-murine leukemia virus, Garaude et al. studied the role of PKCθ in the T cell leukemia progression. They found that the incidence of leukemia was higher in the PKCθ-deficient mice compared with that in the WT mice with the same type of leukemic cell and similar size of spleen and thymus, thus indicating that PKCθ plays an essential role in the immune response to leukemia in mice [30]. Besides, Notch signaling has been found to play an important role in the T-cell differentiation and leukemogenesis by interacting with the pre-T Cell Receptor pathway, which converges on the regulation of distinct NF-κB pathways [31]. Felli et al. found that PKCθ was a downstream target of Notch3 signaling and it could promote the development of the Notch3-induced T-cell leukemogenesis [32]. 

The graft-versus-leukemia (GVL) effect after allogeneic bone marrow transplantation (BMT) represents a powerful immunotherapy to cure hematopoietic malignancies, such as leukemia since the graft contains donor T cells that can eliminate residual malignant cells [33]. However, the donor T cells also recognize the histocompatibility antigens of the healthy cells in the recipient and mount an immune attack against the recipient cells, thereby damaging the recipient organs. This potentially severe consequence of BMT that attacks the recipient body is called the graft-versus-host disease (GVHD). Besides, the commonly used agents that could limit GVDH by preventing the T cells activation are broadly immunosuppressive and can lead to the life-threatening infections. Thus, finding a therapeutic target to inhibit GVHD while maintaining the GVL effect as well as the immune response to the antigen at the same time is very important [34,35]. Using mouse models, Valenzuela et al. found that T cells lacking PKCθ were unable to undergo a strong expansion and cause damage to the recipient organs, while these T cells still maintain both GVL and virus infection responses after BMT [35]. These findings suggested that PKCθ could be a potential therapeutic target that is required for deleterious but not beneficial functions of donor T cells after BMT. Using genetic and pharmacologic approaches, Haarberg et al. confirmed that targeting PKCθ, together with PKCα, by using pharmacological inhibitors would be a good therapeutic strategy to inhibit GVHD while preserving functional GVL immune responses after BMT [36]. Concerning B-cell and T-cell acute lymphoblastic leukemia (ALL), different therapeutic approaches have been developed to cure these two subtypes of ALL. It is therefore important to identify markers that could rapidly distinguish these two types of ALL in order to apply the appropriate treatment. For this purpose, Ma et al. showed that PKCθ and CD3D (T-cell surface glycoprotein CD3 delta chain) together could help to discriminate the B-cell and T-cell ALL, while a single gene could not distinguish the two ALL subtypes effectively [37].

### 3.2. Gastrointestinal Stromal Tumor 

Gastrointestinal stromal tumors (GIST) are the most common type of mesenchymal tumors of the gastrointestinal tract. GISTs harbor the gain-of-function mutations in the *KIT* oncogene, which leads to the overexpression of activated mutant KIT receptor tyrosine kinase. The overexpression of KIT plays an essential role in the development of GIST. The expression of KIT is regarded as the gold standard for the diagnosis of GIST based on histological and immunohistochemical method [38]. However, a small set of GIST (about 5%) is KIT-negative, which causes diagnosis difficulties. Besides, imatinib, an inhibitor of KIT/PDGFRA kinase, is a standard therapy used to cure metastatic GIST; however, the low expression of c-kit oncoprotein in some GISTs could lead to the development of a therapeutic resistance [39]. Thus, finding a new marker to develop a highly specific and sensitive diagnostic tool and effective therapeutic strategy is an urgent need. Blay et al. analyzed the transcriptional profiling data of sarcomas and found that PKCθ was highly and specifically expressed in GIST but not in other mesenchymal or epithelial tumors, including non-GIST KIT-positive tumors [40]. Using immunohistochemistry to detect the expression of PKCθ and KIT in 48 samples of GIST, Motegi et al. found that PKCθ expression was positive in 85% of GISTs and was positive in all KIT-negative GISTs (6 samples), thus revealing that PKCθ could be a useful marker for GIST with negative-KIT [41]. The PKCθ antibody (clone mAb 27, BD Bioscience) used in this study is specific to PKCθ, since it recognizes the C2-like domain of human PKCθ and does not cross-reacts with others PKC isozymes. This antibody has been used in the following studies detecting PKCθ in GIST and in breast tumors (see Section 3.3). In order to evaluate the utility of PKCθ as a diagnostic marker, Kim et al. characterized the expression levels of PKCθ, KIT, CD34, α-smooth muscle actin and S-100 protein by immunohistochemistry in a larger number of GISTs (220 samples) and found that almost all the cases were PKCθ-positive or KIT-positive, which indicates that PKCθ could be an important diagnostic marker of GIST [42]. Zhu et al. studied the signaling pathways that are dependent on the KIT activation in GIST and found that PKCθ interacted with KIT in co-immunoprecipitation experiment [43]. Their data also indicated that the tyrosine phosphorylation of PKCθ was KIT-dependent in GIST [43]. By studying the effects of shRNA-mediated PKCθ knockdown in GIST cell lines, the same group found that PKCθ regulated the KIT expression in both imatinib-sensitive and imatinib-resistant GIST cell lines. PKCθ knockdown also led to a prominent decrease in the AKT activation and to a reduction of GIST cell proliferation. Consistent with the reduced cell growth, PKCθ knockdown resulted in an upregulation of cyclin-dependent kinase inhibitors p21 and p27 expression and a downregulation of cyclin A expression [44]. In a later study, Ou et al. found that PKCθ, JUN and the Hippo pathway coordinately regulated the cyclin D1 expression in GIST [45]. Lately, a work from Kim et al. have explored the relationship between PKCθ and KIT expression by analyzing the expression of the mutant KIT protein that is the one overexpressed in GIST. They found that PKCθ mediated the stabilization of mutant KIT protein by increasing its retention in the Golgi [46].

### 3.3. Breast Cancers

We and other groups found that PKCθ was highly expressed in estrogen receptor negative (ER−) human breast tumors at transcript [47,48] and protein levels [49], whereas it was not expressed in normal breast epithelia and weakly expressed in ER+ breast tumors. The ER− breast cancers usually belong to the triple negative breast cancers (TNBC) since the epithelial breast cancer cells, which do not express the estrogen receptor, often do not express the progesterone receptor and the Her2 protein. The TNBC are more invasive tumors compared to the ER+ ones. In addition to tumor samples, we also found that PKCθ was also highly expressed in the TNBC cell lines compared to the ER+ ones and it is present under its active form as reflected by the expression of phosphoThr-538 PKCθ [48,50]. Our group investigated the cytoplasmic role of PKCθ in cancer cells and we showed that PKCθ was implicated in the mammary tumorigenesis induced by c-Rel, a member of NF-κB family. We discovered a novel NF-κB activation pathway by PKCθ. More precisely, we found that a high level of PKCθ inhibited ERα expression through the activation of the Akt/FOXO3A pathway. Then, this inhibition resulted in the increase of c-Rel activity and to the transcription of c-Rel target genes (implicated in growth and survival such as c-Myc), thereby leading to tumorigenesis [48]. Consistently, we also found that PKCθ increased the expression of RelB (unpublished data), another NF-κB family member that inhibits ERα expression [51] and that promotes mesenchymal phenotype of breast cancer cells [52]. Later, we uncovered another role of cytoplasmic PKCθ and reported that PKCθ induced the migration and invasion of breast cancer cells. Mechanistically, PKCθ activation increased the phosphorylations of FRA-1, a member of AP-1 family. These PKCθ-mediated phosphorylations increased the protein stability and transcriptional activity of FRA-1, thereby inducing the expression of FRA-1 target genes (implicated in cell migration and invasion such as MMP-1) [50,53]. We showed that PKCθ did not directly phosphorylate the FRA-1 protein but acted through the ERK and SPAK pathways to stabilize the FRA-1 protein [50]. In addition to its localization in the cytoplasm, PKCθ in its active form (phosphoThr-538 PKCθ) was also found to distribute in the nucleus of TNBC cells [54]. However, while the data from Zafar et al. indicated that PKCθ exclusively localized into the nucleus of TNBC cells [54], in our hand, PKCθ localization in TNBC cells appeared to be mainly cytoplasmic with a small fraction present in the nucleus (unpublished data). This discrepancy may be due to the subcellular fractionation methods (not described in Zafar et al. publication) and/or the PKCθ antibodies used by both groups (sc-212, Santa Cruz antibody used by Zafar et al.; and E1I7Y, Cell Signaling Technology and clone mAb 27, BD Bioscience antibodies used by our group). Supporting the idea of an active role for PKCθ in the nucleus, a previous report showed that phosphoinositide signaling that generates DAG, a PKCθ co-activator, appeared to be intact in the nucleus [55]. In breast cancer cells, Zafar et al. found that chromatin-bound-PKCθ was part of an active transcription complex that controlled the expression of genes involved in the epithelial to mesenchymal transition (EMT) and cancer stem cell [54,56]. Through this transcriptional control, PKCθ promoted EMT, a key process implicated in the initiation of metastasis in cancer. Sutcliffe et al. described that Chromatin-bound-PKCθ negatively regulated microRNAs transcription, such as miR-200c and miR-9, in T cells [24]. These microRNAs are key factors in tumor metastasis [57] suggesting that they could also play a relevant role in EMT process in invasive breast cancer cells with high level of PKCθ. Later, Boulding et al. found that Chromatin-bound-PKCθ phosphorylated Lysine-specific demethylase 1 (LSD1) to regulate its demethylase activity and promote LSD1 effect on EMT [58]. Altogether, these studies showed that PKCθ was implicated in various steps of breast cancer progression such as proliferation, migration and invasion. Thus, targeting PKCθ could be a good combinatorial therapeutic approach since TNBCs are unresponsive to hormonotherapy and often become resistant to chemotherapy. In fact, PKCθ expression was reported to be increased in MCF7/ADR cells, a MCF7 breast cancer cell line that acquired resistance against the chemotherapeutic agent, adriamycin [59]. More recently, PKCθ has been shown to lower apoptosis induced by chemotherapeutic agents (doxorubicin/adriamycin and paclitaxel) [60]. Byerly et al. found that inhibiting PKCθ kinase activity in TNBC cells enhanced the chemotherapy sensitivity by modulating the expression of Bim, a pro-apoptotic BCL2-family protein [60] and PKCθ knockdown reduced tumor growth of TNBC cells [61]. To conclude, cytoplasmic and nuclear PKCθ are involved in breast cancer progression and chemoresistance.

### 3.4. Other Cancers

In a mouse model of lung cancer, the adaptor protein, TBK-binding protein 1 (TBKBP1), has been shown to mediate PKCθ direct phosphorylation of TANK-binding kinase 1 (TBK1) in response to epidermal growth factor (EGF) stimulation, leading to the activation of the oncogenic activity of TBK1. TBK1 was also responsible for the EGF induction of PD1 ligand 1 (PD-L1), a protein implicated in immunosuppression. Altogether, these data described a PKCθ-TBKBP1-TBK1 signaling pathway that would control tumor growth and immunosuppression [62]. In renal cancer cells, the natural compounds, englerin A and tonantzitlolone, were reported to activate PKCθ and act as an anti-tumor agent in a PKCθ-dependent manner. By activating PKCθ, these compounds induced an insulin resistant phenotype in tumor cells through the activation of the heat shock factor 1 (HSF1) [63,64]. PKCθ has also been proposed to phosphorylate and inhibit the activity of the Gα-interacting, vesicle-associated protein (GIV) in Hela cells, thereby inhibiting Akt activation, actin stress fiber formation and cell migration [65]. Chu et al. found an aberrant expression of PKCθ in oral squamous cell carcinoma and their data suggested that high nuclear PKCθ expression could correlate with disease recurrence and poor survival [66]. In ovarian cancer, phosphorylated PKCθ (Ser-695) has been detected at the protein level, however its function is unknown in this type of cancer [67]. Cytoplasmic PKCθ was also expressed in Ewing sarcoma/primitive neuroectodermal tumor (ES/PNETs) and in malignant peripheral nerve sheath tumors (MPNSTS). Specific “dot-like” pattern of PKCθ staining could be useful for the diagnosis in ES/PNETs, although the role of PKCθ remains unknown in both ES/PNETs and MPNSTS [68]. Exploring PKCθ expression in other cancers might be interesting and deciphering the function of this protein in all these cancers could be important to understand its pathological function.

## 4. Perspectives

As summarized in Table 1, the function and mode of action of PKCθ are different depending on the type of cancer. However, in most of these cancers, the presence of a strong PKCθ level leads to the abnormal proliferation, migration and invasion of tumor cells, thereby promoting tumor aggressiveness. The PKCθ-mediated control of the tumor aggressiveness is somehow different from the PKCθ-mediated physiological control of the immune response (Figure 2). However, these two controls share some similarities. Indeed, PKCθ can activate AP-1 and NF-κB transcription factors in both tumor cells and T cells but with the implication of different AP-1 and NF-κB subunits. Moreover, PKCθ-mediated control can occur either in the cytoplasm or in the nucleus of both tumor cells and T cells.

The challenge for cancer researchers is now to further clarify whether PKCθ tumoral function is dependent on its kinase activity and to clearly define PKCθ as a new therapeutic target for human cancers such as GIST and aggressive breast cancers. The use of specific PKCθ inhibitors may provide a new cancer therapy strategy for these types of cancer that are difficult to treat. Considering that the PKCθ knock-out mice are generally healthy and fertile, we would expect that the use of specific PKCθ inhibitors as cancer therapy would inhibit the proliferative and invasive ability of this kinase while producing minimal toxic side effects [9,10]. Moreover, the role of PKCθ in the immune system could provide additional benefit to such cancer therapy. Indeed, the immune system is involved in the control of tumor progression at two opposite directions: on the one hand, it possesses an inhibitory effect on the development of the tumor via the activation of Th1 responses that are PKCθ-independent and on the other hand, it could promote the tumor invasion by activating the Th2 responses that are PKCθ-dependent [69]. Thus, the use of specific inhibitors could not only block the PKCθ intrinsic function in tumor cells but could also suppress the Th2-driven pro-invasive function of the microenvironment.

## Figures and Tables

**Figure 1 biomolecules-11-00221-f001:**
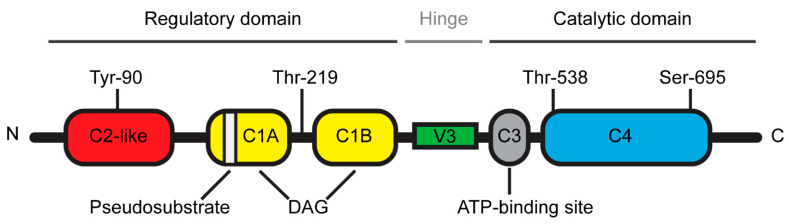
Schematic representation of Protein Kinase C theta (PKCθ) structure.

**Figure 2 biomolecules-11-00221-f002:**
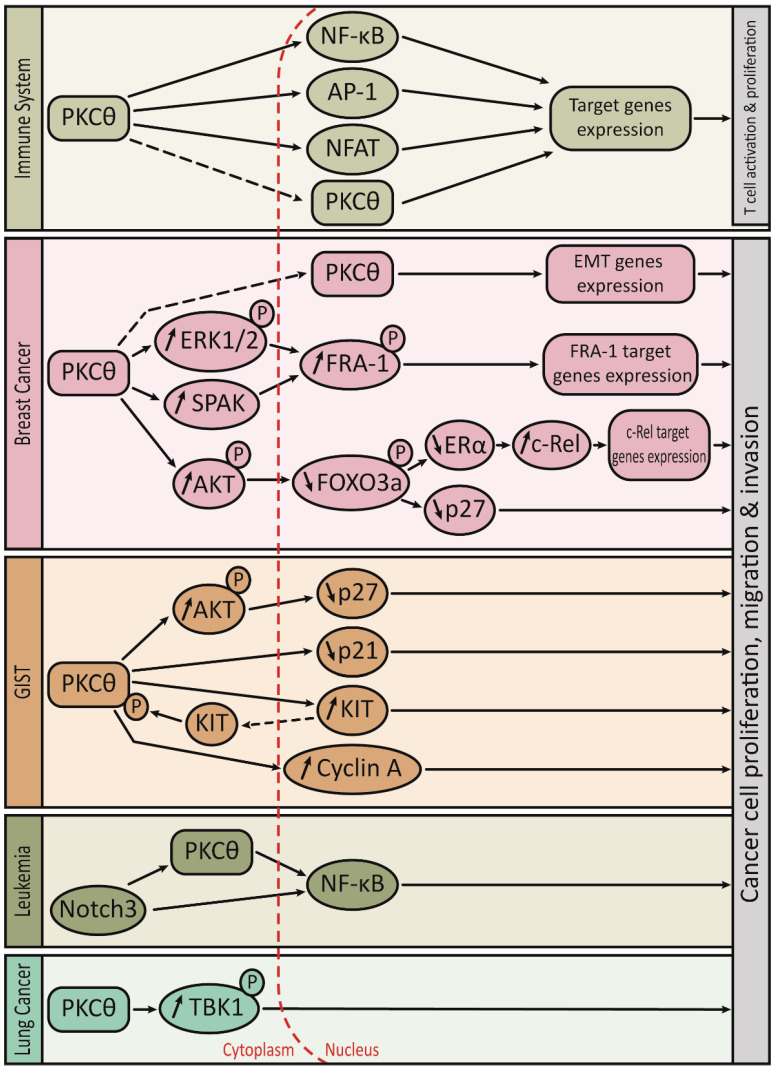
Cartoon summarizing the PKCθ-mediated signaling pathways controlling the immune response and the tumor aggressiveness.

**Table 1 biomolecules-11-00221-t001:** Summary of PKCθ function and mode of action in various cancer types.

Cancer Type	Biological Function	Mechanism of Action	Functional Localization	Ref.
Leukemia	Leukemogenesis	Activation of Notch3-PKCθ pathway	Membrane	[32]
		↗ NF-κB activity		
	Immunosuppression	Unknown	Unknown	[30]
	Biomarker	Sort between B-ALL and T-ALL	-	[37]
GIST	Proliferation	↗ KIT and CyclinA; ↘ p27 and p21	Cytoplasmic	[44,45,46]
	Biomarker	Diagnosis of KIT-negative GIST	-	[40,41]
Breast Cancer	Tumorigenesis	↗ c-Rel activity	Cytoplasmic	[48]
		Activation of AKT/FOXO3a pathway		
	Migration & Invasion	↗ FRA-1 protein stability and activity	Cytoplasmic	[50,53]
		Activation of ERK and SPAK pathways		
		Activation of EMT genes transcription	Nuclear	[54,56,58]
	Potential biomarker	High PKCθ expression in TNBC	-	[47,48,49]
	Tumor growth	Unknown	-	[61]
	Chemosensitivity	↘ Bim	-	[60]
Lung Cancer	Tumorigenesis	Activation of PKCθ/TBKBP1/TBK1 pathway	Cytoplasmic	[62]
	Immunosuppression	PKCθ/TBKBP1/TBK1 pathway facilitates EGF induced PD-L1 expression	Cytoplasmic	[62]
Renal Cancer	Anti-tumor & insulin resistance	Activation of HSF1	Cytoplasmic	[63,64]
Oral squamous cell carcinoma	Diagnosis marker	High nuclear PKCθ level	Nuclear	[66]
Ewing Sarcoma	Potential biomarker	High PKCθ expression	-	[68]
Ovarian Cancer	Unknown	Strong phospho-Ser695-PKCθ level	-	[67]

B-ALL: B-cell acute lymphoblastic leukemia. T-ALL: T-cell acute lymphoblastic leukemia. TNBC: Triple Negative Breast Cancer.

## Data Availability

Not applicable.

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
