# Peer review of "The Emerging Function of PKCtheta in Cancer"

_biomolecules, 2021, doi:10.3390/biom11020221_

Round 1

Reviewer 1 Report

This manuscripts attempts to cover the potential link between PKC-theta and cancer. Unfortunately, the paper is poorly written and organized, and section 2 of the manuscript focusing on the role of PKC-theta in T cell activation is full of inaccuracies and failure to cite relevant references, likely reflecting the fact that the authors have little understanding of T cell biology and activation. Rather than describing in detail all the deficiencies in the manuscript, I am providing just a few glaring examples:

1) Line 55: Failure to cite the two papers that reported the characterization of PKCtheta-deficient mice

2) Lines 56-57: Some studies report a defect in Th1 responses in PKCtheta-deficient mice, particularly in vivo.

3) Line 58: T cells do NOT recognize APCs. They recognize a peptide-MHC complex presented by APCs.

4) Line 59: Ref. #7 is wrong. The authors should instead cite the original papers that described the IS.

5) The authors fail to cite a paper showing that the catalytic activity of PKC-theta is also required for its translocation to the IS

6) Lines 77-78: IL-2 is NOT required for T cell activation. It is required for T cell proliferation once they have been activated.

7) Lines 85-87: Disruption of PKC-theta does NOT lead to the development of Th2- and Th17-mediated autoimmunity. The opposite is true because PKCtheta-deficient mice are resistant to these diseases. 

8) The sentence in lines 105-107 is poorly written; not clear what the authors are trying to say

9) Failure to define acronyms, e.g., CD3D (line 111), TBKBP1 (line 189), TBK1 (line 190).

Reviewer 2 Report

Nicolle et al present a review of protein kinase C theta (PKCθ) in cancer. They discuss the implications of PKCθ in various cancer types, notably leukemia, gastrointestinal stromal tumors and breast cancer, and PKCθ-mediated signaling events involved in cancer development and progression. They posit that PKCθ inhibitors could be used in the treatment of cancers that involve PKCθ.

There are several reviews on the broader subject of protein kinase C isoenzymes as potential targets for cancer therapy, including Hofmann (2004), Kang (2014), Cooke et al (2017), Isakov (2018), and the outstanding Parker et al (2020). A focus on PKCθ is welcome.

The majority of the references cited by Nicolle et al are quite old, with only 8 of 52 published in the last three years, which might dilute its contribution to the field. Whilst the manuscript is reasonably written, it would benefit from extensive editing for the English language (see below).

A graphical abstract or figures summarising the pertinent points may be beneficial.

The title of subsection 2.3, "Related diseases", is ambiguous and should be edited to be more informative. A more detailed review of PKCθ inhibitors beyond the citing of Zhang et al 2013, would enhance the impact of the manuscript. For example, the PKCθ inhibitor AEB071 has been reported to attenuate the stemness of gastric cancer cells (Yuan et al 2019). Are there other novel inhibitors of PKCθ?

What are the outcomes from clinical trials of PKCθ inhibitors? Are they safe and effective? What are the challenges, including drug specificity, timing, patient subsets, etc?

The review of the cancer literature could be more comprehensive. A brief search suggested that several papers were not cited, including but not limited to:
Boulding et al 2019 DOI: 10.1038/s41598-017-17913-x
Kang et al 2010 DOI: 10.4143/crt.2010.42.3.135
Zhu et al 2007 DOI: 10.1038/sj.onc.1210464
Gill et al 2001 DOI: 10.1046/j.1432-1327.2001.02326.x

There was no mention of PKCθ in melanoma.

Extensive editing for English language is recommended, including but not limited to the following:
Line 13: "plays critical role in T cell" to "plays critical roles in T cell".
Line 14: "more and more evidences" to "increasingly more evidence".
Line 19: "conducting to malignant phenotype" to "leading in a malignant phenotype".
Line 32: "homoeostasis" to "homeostasis".
Line 33: "In the context of immune system" to "In the context of the immune system".
Line 36: "growing evidences implicate" to "growing evidence implicates".
Line 43" "approximatively" to "approximately".
Line 47: "substrate binding" to "substrate-binding".
Line 54: "The generation and analysis of PKCθdeficient mice has unraveled" to "The generation and analysis of PKCθdeficient mice have unraveled".
Line 59: "called immunological synapse" to "called the immunological synapse".
Line 64: "participates into the" to "participates in the".
Line 67: "involved in the specific localization of PKCθ and in the organization of" to "involved in the specific localization of PKCθ, and in the organization of".
Line 81: "PKCθ is able to enter the nucleus of activated T cells to directly binds to the chromatin in order to regulate the expression of immune response genes and of microRNAs involved in cytokine regulation" to "PKCθ can enter the nucleus of activated T cells to directly bind to the chromatin to regulate the expression of immune response genes and microRNAs involved in cytokine regulation".
Line 89: "the therapeutic use of specific PKCθ inhibitors provide an interesting approach" to "the therapeutic use of specific PKCθ inhibitors provides an interesting approach".
Line 91: "show encouraging results in the context of immunosuppressive therapy for autoimmune disease such as psoriasis and for organ transplantation" to "show encouraging results in the context of immunosuppressive therapy for autoimmune diseases such as psoriasis and organ transplantation".
Line 96: "moloney-murine" to "Moloney-murine".
Line 99: "with same" should be "with the same".
Line 99: "which indicate that" to "which indicates that".
Line 101: "interacting with pre-T Cell" to "interacting with the pre-T Cell".
Line 103: "promote development of" to "promote the development of".
Line 105: "Allogeneic Bone Marrow transplant" to "Allogeneic bone marrow transplant".
Line 106: "eliminate residual tumor cells through immune system" to "eliminate residual tumor cells through the immune system".
Line 113: "while single gene" to "while single a gene".
Line 115: "Gastrointestinal stromal tumor (GIST) is a common type of mesenchymal tumor, and the KIT receptor tyrosine kinase encoded by the KIT oncogene plays an essential role in the development of GIST." to "Gastrointestinal stromal tumors (GISTs) are the most common mesenchymal tumor of the gastrointestinal tract. Most GISTs harbour gain-of-function mutations in the receptor tyrosine kinase encoded by the KIT oncogene plays an essential role in the development of GISTs".
Line 117: "GISTs harbor a gain-of-function mutations" to "GISTs harbor gain-of-function mutations".
Line 118: "gold standard for diagnosis" to "gold standard for the diagnosis".
Line 146: "PKCθ has been shown to be expressed in estrogen-negative (ER-) tumors at transcript [34] and protein levels [35], whereas it is not expressed in normal breast tissues" to "PKCθ is expressed in estrogen-negative (ER-) tumors at transcript [34] and protein levels [35], whereas it is not expressed in normal breast tissues".
Line 152: "was investigated in cancer cell and" to "was investigated in cancer cells and".
Line 154: "High level of" to "A high level of".
Line 171: "in the nucleus, previous report showed that" to "in the nucleus, a previous report showed that".
Line 174: "Similarly to its role in T cells" to "Similar to its role in T cells".
Line 175: "and cancer stems cell inducible genes" to "and cancer stem cell inducible genes".
Line 176: "in the initiation of metastasis in Cancer" to "in the initiation of metastasis in cancer"
Line 176: "Chromatin-bound-PKCθ also negatively regulate microRNAs transcription in T cells such as miR-200c and miR-9" to "Chromatin-bound-PKCθ also negatively regulates transcription of microRNAs, such as miR-200c and miR-9, in T cells".
Line 177: "These microRNAs are key factors in tumor metastasis [44] suggesting that they could also play a relevant role in EMT process in high level PKCθ breast cancer cells" to "These microRNAs are a key factor in tumor metastasis [44] suggesting that they could also play a relevant role in the EMT process in breast cancer cells with high levels of PKCθ."
Line 181: "a good complementary therapeutic approach" to "a good combinatorial therapeutic approach".
Line 182: "hormonotherapy" to "hormone therapy".
Line 184: "in TNBC cells enhance chemotherapy sensibility" to "in TNBC cells enhances chemotherapy sensibility".
Line 186: "are involved in several breast cancerous processes and in chemoresistance" to "are involved in breast cancer progression and chemoresistance".
Line 189: "TBKBP1 adaptator protein" to "TBKBP1 adaptor protein".
Line 192: "It suggests that PKCθ-TBKBP1-TBK1 signaling pathway" to "This suggests that the PKCθ-TBKBP1-TBK1 signaling pathway".
Line 194: "at protein levels, however its function is unknown" to "at the protein level, however, its function is unknown".
Line 219: "As summarized in the Table 1" to "As summarized in Table 1".
Line 220: Do you mean "this control occurs either at the nuclear or at the cytoplasmic level" or "this control occurs at both the nuclear and the cytoplasmic level"?
Line 222: "as a new therapeutic target for human cancers such as GIST and breast cancers" to "as a new therapeutic target for human cancers such as GIST and breast".
Line 223: "If both are confirmed, this will provide new cancer therapy strategy" to "If both are confirmed, this may provide a new cancer therapy strategy".
Lin 229: "Indeed, the immune system is involved in control of tumor progression" to "Indeed, the immune system is involved in the control of tumor progression".
Line 230: Do you mean "on the one hand, it possesses an inhibitory effect on the development of the tumor via the activation of Th1 responses that are PKCθ-independent and on the other hand, it could promote the tumor invasion by activating the Th2 responses that are PKCθ-dependent" or "on the one hand, it possesses an inhibitory effect on tumor development and progression via the activation of Th1 responses that are PKCθ-independent, and on the other hand, it could promote the tumor invasion by activating the PKCθ-dependent Th2 responses"?

Reviewer 3 Report

The work b Nicolle et al. entitled "The emerging function of PKCtheta in cancer" describes the role of the novel PKCθ isozyme in different stages of cancer development, mainly related to its role in the immune system and immunological synapsis. The review article is interesting for the field, it is novel and well organized and written. Moderate revisions are required:

Major points

  1. A more specific subtitle in paragraph 2.2 may give a better idea about the paragraph content: i.e. Role of PKCθ in the immune system. Any sentence that describes more precisely the contents of the paragraph
  2. In line 107 authors explain "finding a therapeutic target to inhibit the graft-versus-host disease (GVHD and maintain the GVL as well as the immune response to the antigen at the same time is ver important".  In what sense is it important?. Is it critical in order to find more efficient therapeutic drugs? Is it critical because there is currently no efficient drugs with such effect?. Authors may need to clarify this
  3. In this same paragraph lines 105-111, the idea of PKCθ as the target to inhibit GVHD while maintaining GVL is not complete clear. Do authors think this protein would be a good target protein for the development of such kinds of drugs? How is it related to the fact that PKCθ is expressed is B- and T-cell subtypes of ALL. Are these two ideas related?
  4. On line 129 and below, what kind of antibodies do Motegi et al. use to detect PKCθ. This may be a crucial fact because of the similarities between all PKC isozymes. Is the antibody used in the work of Motegi et al specific of PKCθ?
  5. Lines 146-149, in breast cancer tumors, what kind of cells express PKCθ. Are they immune cells, other cell types? 
  6. In lines 166-169 authors discuss the differences between their data and the data from Zafar et al regarding the specific nuclear expression of PKCθ. Is there any reason for the discrepancy? May it be explained by the antibodies used? It would be interesting to know the antibodies used by each group to understand the specificity of such antibodies, specially when the work they cite is unpublished. Could thre be any cross reaction that explains the presence of PKCθ in the cytoplasm of TNBC cells? A short explanation may be useful
  7. A drawing with the different signaling pathways involving PKC in immune cells and tumor cells may be helpful at understanding the role of this kinase in the different cell types and cancers

Minor points:

minor spelling errors may be found i.e: line 82 instead of "to directly binds" it should be "to directly bind", authors may want to check the whole manuscript.

Round 2

Reviewer 2 Report

All of my comments have been addressed. There appears to be some variation in the font size, otherwise accept in its current form.

Reviewer 3 Report

Authors have given a response to all comments. I believe the manuscript is now ready for publication